# Decreased continuous sitting time increases heart rate variability in patients with cardiovascular risk factors

Natsuki Nakayama[1]*, Masahiko Miyachi[2], Koji Tamakoshi[1], Toshio Hayashi[1], Koji Negi[3], Koji Watanabe[4], Makoto Hirai[1,5]

1 Nagoya University Graduate School of Medicine, Higashi-ku, Nagoya, Japan, 2 Chutoen General Medical Center, Kakegawa, Shizuoka, Japan, 3 Negi Clinic, Inazawa, Aichi, Japan, 4 Watanabe Medical Clinic, Taketoyo, Chita, Aichi, Japan, 5 Sugiyama Jogakuen University, Chikusa-ku, Nagoya, Japan

* nakayama@met.nagoya-u.ac.jp

**Data Availability Statement:** All relevant data are within the paper and its Supporting Information files.

## Abstract

### Aim

The purpose of the present study was to elucidate the relationship between high-frequency heart rate variability (HF HRV) and continuous daytime sitting time in patients with cardiovascular risk factors such as mild hypertension and/or stable angina pectoris.

### Background

Decreased HF HRV precedes the progression and worsening of cardiovascular diseases. Continuous sitting behavior is a major risk factor for developing metabolic syndrome and is associated with cardiovascular disease, diabetes mellitus, renal failure, sarcopenia and osteoporosis. Risk factors for cardiovascular disease can be affected by continuous daytime sitting behaviors.

### Design

The present study design was a post-hoc comparison.

### Methods

Patients treated at two different primary care clinics from 2014 to 2018 were enrolled in this study (n = 53). We assessed HF HRV and continuous sitting time using 24-hour Holter electrocardiography and an activity meter at baseline and 6 months. HF HRV was calculated during sleep.

### Results

Sitting time had decreased in 22 patients (decreased group) and increased in 31 patients (increased group) after 6 months. The mean patient ages were 73.1 and 72.0 years in the decreased and increased sitting time groups, respectively (p = 0.503). HF HRV during sleep had increased after 6 months in the decreased sitting time group. Compared with the increased group, the decreased group showed significantly higher HF HRV during sleep

**Funding:** This work was supported in part by the grants-in-aid from the Japanese Ministry of Education, Culture, Sports, Science and Technology, Tokyo, Japan (Grants-in-aid for Nakayama, #23792624). The funders had no role in study design, data collection and analysis, decision to publish, or preparation of the manuscript. There was no additional external funding received for this study.

**Competing interests:** The authors have declared that no competing interests exist.

after 6 months by two-way repeated-measures ANOVA after adjustment for age, sex and change in activity (p = 0.045).

## Conclusion

These results suggest that a decrease in sitting time might induce parasympathetic activity during sleep. Therefore, reducing continuous sitting time during the day might contribute, in part, to improving the prognosis of patients with cardiovascular risk factors not only by avoiding muscle loss but also by providing positive influences on parasympathetic tone during sleep.

## Introduction

Heart rate variability (HRV) has been analyzed using the normal R-R waveform on electrocardiography and applied to a wide range of conditions, including cardiovascular disease risk factors, such as hypertension and diabetes [1, 2]. Parasympathetic nervous activity is measured as high-frequency (HF) HRV, and decreased HF HRV has been shown to precede disease progression and worsening [2, 3]. Lower HF HRV is associated with a higher risk of fatal and non-fatal cardiovascular disease [3]. For example, lifestyle habits, such as napping, have been reported to increase the HF HRV index in patients with cardiovascular risk factors [4]. The data obtained by Nakayama et al. [5] demonstrated that increased physical activity and low-intensity physical activity were associated with increased HF HRV in patients with cardiovascular risk factors. Moreover, increased physical activity has been shown to increase HF HRV during the first hour after sleep onset [6]. Many previous studies have reported that the management of overall lifestyle habits, such as physical activity, is important for reducing the risk of cardiovascular disease and risk factors, such as hypertension and dyslipidemia [7–9]. Furthermore, lifestyle habits involving a tendency to sit lead to a reduction in muscle strength and increase in the risk of developing many diseases including cardiovascular diseases [10–12]. In addition, it has been reported that low physical activity levels and sedentary lifestyles that involve sitting for a long period of time increase the risk of plaque formation and subsequent stenosis in patients with atherosclerosis [13, 14]. Sedentary lifestyle behaviors, such as watching TV for a long time, reportedly increase the predisposition to develop obesity and may be related to symptoms of high blood pressure and musculoskeletal disorders. A sedentary lifestyle accelerates biological ageing, is a major risk factor for developing metabolic syndrome and is associated with cardiovascular disease, diabetes mellitus, renal failure, sarcopenia and osteoporosis [15]. Therefore, increased physical activity during leisure time may improve outcomes related to the genetic association with cardiovascular risk factors [16, 17]. In a study of blue-collar workers, more sitting time during leisure time tended to decrease HRV, and more standing time at work increased HF HRV [18]. These previous studies have suggested that risk factors for cardiovascular disease are likely affected by lifestyle habits in senior citizens and workers who often remain seated. However, the impact of continuous sitting time (ST) during the daytime on autonomic nervous activity has not been fully clarified. The purpose of the present study was to elucidate the relationship between HF HRV during sleep and continuous ST during the day in patients with mild hypertension and/or stable angina pectoris.

## Methods

### Design

The design of the present study was a post-hoc comparison.

## Participants

Patients with mild hypertension and/or stable angina pectoris who were treated at two different primary care clinics from 2014 to 2018 were enrolled in this study. In the present study, all patients provided complete details on their lifestyles and had visited outpatient clinics for 6 consecutive months (n = 53). Participants had no cardiovascular events such as myocardial infarction or stroke before December 2019. The following patients were excluded from this study: (1) patients with a pacemaker; (2) patients with transient or continuous atrial fibrillation; and (3) patients with a sleep disorder or with complaints of sleep disorder at the time of data recording.

## Data collection

Patients were followed for 6 months from the start of the study. We met with patients once a month for the 6 months of follow-up to consult on the importance of reviewing lifestyle behaviors and to discuss the risks of continuous long periods of sitting. The amounts of physical activity were measured for 30 days at baseline and after 6 months (Active Style Pro HJ-350IT, Omron Colin Co., Ltd. Tokyo, Japan). In addition, Holter 24-hour electrocardiography was evaluated twice: once at baseline and once after 6 months (EC-2H 2-Channel Holter ECG System; Labtech Ltd. Hungary).

## Sitting time (ST)

The patients were required to answer whether they had continuous sitting periods of approximately 60 min during the day except for naps. In other words, the patient was asked to state the amount of time spent continuously watching TV, reading, using the internet, etc. every day. This question was the same as that used in previous studies [12, 17, 19]. The patients kept a log of their daily living activities while wearing a 24-hour ambulatory Holter electrocardiographic monitor. In addition, they continuously wore an activity meter on their waists using a belt, except while bathing. With patients wearing a waist-mounted activity meter, we were able to identify the times they moved their waist and thus determine whether the patients were active or sedentary. We calculated all data as calories burned during physical activity according to the height and weight of each subject. We determined the amount of physical activity per activity meter-wearing time (min). All records from each activity meter were reviewed on a computer by the authors. For the final analysis of ST change, we assessed the time spent sitting on the couch, watching television, reading a book, and using the internet based on the patient's log, physical activity from the activity meter and heart rate data from the Holter 24-hour ambulatory electrocardiograms.

## HRV measurement

We confirmed all 24-hour electrocardiogram data using a computer and extracted only the normal heart rate. Furthermore, we confirmed the sleeping and waking time using the patient's log and changes in the heart rate. In addition, we used the maximum entropy method for the frequency analysis (MemCalc/Win2; GMS Co., Ltd. Tokyo). The HF HRV component was considered to reflect parasympathetic nerve modulation with a frequency band of 0.15 to 0.4 Hz. The HF HRV index calculated the average during the night. The HF HRV index was logarithmically converted and is displayed as the mean and standard deviation (SD).

## Ethical considerations

We explained the present study to all patients with the documents approved by the ethics committee, and all the participants submitted written informed consult. The present study was conducted with the approval of the Bioethics Review Committee at Nagoya University (260024–1, 201700953).

## Statistical analysis

We performed all statistical analyses using commercially available software packages (SPSS ver. 26J Regression for Windows, SPSS Inc. Tokyo, Japan). We used chi-square tests and t-tests for between-group comparison. In addition, two-way repeated-measures ANOVA was used to compare log HF HRV values during sleep between the increase and decrease groups after 6 months. Statistical significance was indicated by p values $< 0.05$.

## Results

Among 72 patients, 19 patients had difficulty performing Holter 24-hour electrocardiography and/or measuring their physical activity. Fifty-three patients finally completed the study protocol, and we analyzed the data for these 53 patients. These patients were taking medications during the present study, but their medications were not changed throughout the present study period. The baseline characteristics of the population in the study are shown in Table 1. We determined the STs at baseline and after 6 months and divided the subjects into two groups: the decreased ST group (n = 22) and the increased ST group (n = 31). In the decreased ST group, ST decreased by a minimum of 5 minutes and a maximum of 135 minutes after 6 months from baseline, and in the increased ST group, ST increased by a minimum of 0 minutes and a maximum of 108 minutes. Fifteen and 16 men participated in the decreased and

**Table 1. Comparison of patient characteristics for the decreased sitting time and increased sitting time groups at baseline and after 6 months.**

| | Decreased sitting time group (n = 22) | Increased sitting time group (n = 31) | *p* value |
|---|---|---|---|
| Male sex, n (%) | 15 (68.1) | 16 (51.6) | 0.268 |
| Age, years, mean±SD | 73.1±6.6 | 72.0±5.9 | 0.503 |
| BMI, kg/m$^2$, mean ± SD | 23.8±4.0 | 23.9±3.6 | 0.949 |
| Dyslipidemia, n (%) | 12 (54.5) | 16 (51.6) | 1.000 |
| Diabetes mellitus, n (%) | 4 (18.1) | 6 (19.3) | 1.000 |
| Medications, n (%) | | | |
| Angiotensin II receptor antagonist | 12 (54.5) | 21 (67.7) | 0.395 |
| Calcium antagonist | 15 (68.1) | 22 (70.9) | 1.000 |
| β-blocker | 5 (22.7) | 7 (22.5) | 1.000 |
| Baseline | | | |
| Sitting time, min | 70.0±26.5 | 31.9 ± 34.5 | <0.0001 |
| Sleeping time, min | 458.9±84.7 | 497.8±84.3 | 0.105 |
| Physical activity, kcal/wearing time (min) | 0.70 ±0.24 | 0.78 ± 0.23 | 0.231 |
| After 6 months | | | |
| Sitting time, min | 25.3±23.7 | 54.2±40.5 | 0.004 |
| Sleeping time, min | 464.7±46.9 | 465.7±61.8 | 0.953 |
| Physical activity, kcal/ wearing time (min) | 0.77±0.25 | 0.80±0.20 | 0.592 |

Notes. Abbreviations: AP, angina pectoris; HT, hypertension.

**Table 2. Two-way ANOVA measurements comparing log HF during sleep between the decreased sitting time and increased sitting time groups at baseline and after 6 months.**

|  | Decreased sitting time group (n = 22) | Increased sitting time group (n = 31) | Sum of squares | Mean squares | F-value | p value |
|---|---|---|---|---|---|---|
| Baseline | 2.30 ± 0.21 | 2.14 ± 0.38 | 0.861 | 0.861 | 4.217 | 0.045 |
| After 6 months | 2.36 ± 0.32 | 2.16 ± 0.37 |  |  |  |  |

increased ST groups, respectively (p = 0.268). The mean ages were 73.1 years in the decreased ST group and 72.0 years in the increased ST group (p = 0.503). The mean body mass index was 23.8 in the decreased ST group and 23.9 in the increased ST group (p = 0.949). There were no significant differences between the decreased and increased ST groups in terms of age or body mass index. No statistically significant differences in sleeping time or physical activity were found between the decreased and increased ST groups at baseline and after 6 months (p = 0.105, 0.231, 0.953 and 0.592, respectively) (Table 1).

The present study showed that the log of HF HRV during sleep was significantly greater after the 6-month observation period in the decreased ST group than in the increased ST group using the two-way repeated-measures ANOVA adjusted for age, sex and change in activity (p = 0.045) (Table 2).

## Discussion

In the present study, a decrease in ST significantly increased HF HRV during sleep compared to an increased or unchanged ST. The findings of the present study suggest that decreasing the continuous ST during the day might contribute to an increase in parasympathetic nerve activity. In a previous study, Hallman et al. [20] showed that longer sitting times at work weakened the autonomic regulation of the heart, regardless of participation in moderate-to-intense physical activity. In addition, sedentary lifestyles were the most common behaviors associated with chronic illnesses such as hypertension, atherosclerosis, diabetes, and obesity [21]. The combination of aerobic and strength training for heart rate variability in sedentary hypertensive women has been reported to show a significant improvement in vagal dominance among HRV parameters [22]. These previous studies demonstrated the disadvantages of continuous ST and the benefits of exercising for sedentary patients but did not show the benefits of frequent interruptions in ST. Therefore, to our knowledge, the present study is the first report to reveal positive influences of reduced ST on the parasympathetic system and sleep quality in patients with cardiovascular diseases from the viewpoint of reduced ST and the autonomic nervous system.

Increasing continuous ST can adversely affect various organs through cytokines released from the muscles and result in frailty and sarcopenia. Shortening the continuous ST leads to improvements in the sedentary lifestyle and suppresses muscle loss. Accordingly, a decrease in continuous ST is recommended for patients with cardiovascular disease. However, there have been no reports regarding the beneficial effects of shortening ST on the autonomic nervous system or sleep in patients with cardiovascular disease. Prolonged ST and a sedentary lifestyle lead to inadequate production and release of cytokines such as myokine due to inadequate stimulation of the endocrine component of muscles. This biological reaction might cause the deterioration of many organs over time. As a result, prolonged sitting can lead to frailty and the development of sarcopenia [23–25]. Sarcopenia was present in one-third of older hospitalized patients with cardiovascular disease and increased their risk of readmission [26]. Physical inactivity and the loss of muscle mass lead to increased visceral fat deposition, generating an

imbalance between anti-inflammatory and pro-inflammatory conditions, supporting the vicious cycle of sarcopenia, causing an increase in fat mass, and promoting the development of cardiovascular complications [27, 28]. Low-grade chronic inflammation affects glucose and lipid metabolism, endothelial dysfunction, cardiovascular disease, and sarcopenia [15, 29–31]. Previous studies showed that if the endocrine role of muscle is not sufficiently stimulated, as is the case in those with long periods of continuous sitting, sarcopenia may develop, ultimately leading to the development of cardiovascular complications. In other words, frequent interruptions in ST reduce the likelihood of developing sarcopenia by stimulating muscles to release cytokines. Previous studies have suggested that reducing sedentary time and introducing frequent breaks might be beneficial for improving body composition in healthy older people [11]. In this study, patients wore an activity meter on their waist. Patients' ST was considered to be interrupted when there were movements in the lumbar region. This study suggested that the introduction of frequent interruptions in long-term sitting periods stimulated parasympathetic activity during sleep.

Previous studies have shown that a significant increase in physical activity increases 24-hour parasympathetic activity [5]. In the present study, even though there was no significant increase in overall physical activity after 6 months, parasympathetic activity was significantly increased in patients with decreased continuous ST. The autonomic nervous system plays a fundamental role in maintaining cell homeostasis and physiological function. The functionality of the heart and skeletal muscles is partially modulated by the sympathetic and parasympathetic branches of the autonomic nervous system both during rest and during exercise [32]. Patients with sarcopenia have higher muscle sympathetic nerve activity than those without sarcopenia and exhibited reduced parasympathetic-associated modulation, suggesting an adverse effect of sarcopenia on cardiac health [33, 34]. These studies suggest that a sedentary lifestyle, with long periods of continuous ST and reduced levels of physical activity, can trigger body composition changes and affect the autonomic nervous system. The present study revealed that a decrease in continuous ST due to frequent interruptions activated parasympathetic nerve activity during sleep. Therefore, patients might be able to prevent body composition changes and activate parasympathetic nerve activity by frequently interrupting their continuous ST.

Despite these findings, the present study has some limitations. Patients who participated in this study had mild hypertension and/or stable angina pectoris. Therefore, the novel findings of this study may need further studies to be applied to patients with severe hypertension or patients with impaired cardiac function. In addition, the findings of the present study were confirmed when patients with a long ST decreased their ST. Therefore, further studies are also needed to determine whether our novel finding of an increase in HF HRV with a decrease in ST could be applicable to patients with an even shorter ST, such as 30 min.

In the present study, we found that interrupting long periods of sitting with the lumbar spine fixed during the day induced parasympathetic activity during sleep. The decrease in HF HRV is related to a risk of both fatal and nonfatal first cardiovascular events [3]. Furthermore, a low HF HRV value indicates a potentially poor prognosis [35]. Therefore, reducing continuous ST during the day might contribute, in part, to improving the prognosis of patients with cardiovascular risk factors not only by avoiding muscle loss but also by providing positive influences on parasympathetic tone during sleep.

## Supporting information

**S1 Table. DATA: These are the values of the parameter among participating patients.** (PDF)

## Acknowledgments

We thank American Journal Experts (www.aje.com) for performing English language editing.

## Author Contributions

**Conceptualization:** Natsuki Nakayama.

**Data curation:** Natsuki Nakayama, Koji Negi, Koji Watanabe.

**Formal analysis:** Natsuki Nakayama, Koji Tamakoshi.

**Funding acquisition:** Natsuki Nakayama.

**Methodology:** Natsuki Nakayama, Masahiko Miyachi, Makoto Hirai.

**Project administration:** Natsuki Nakayama.

**Supervision:** Koji Tamakoshi.

**Writing – original draft:** Natsuki Nakayama, Makoto Hirai.

**Writing – review & editing:** Natsuki Nakayama, Masahiko Miyachi, Toshio Hayashi, Makoto Hirai.

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
