## [Decision Letter · Decision Letter 0]

29 Apr 2021

PONE-D-21-08345

Decreased Continuous Sitting Time Increases Heart Rate Variability in Patients with Cardiovascular Risk Factors

PLOS ONE

Dear Dr. Natsuki Nakayama,

Thank you for submitting your manuscript to PLOS ONE. After careful consideration, we feel that it has merit but does not fully meet PLOS ONE’s publication criteria as it currently stands. Therefore, we invite you to submit a revised version of the manuscript that addresses the points raised during the review process.

We look forward to receiving your revised manuscript.

Kind regards,

Sharon Mary Brownie

Academic Editor

PLOS ONE

Journal Requirements:

3. Please provide a sample size and power calculation in the Methods, or discuss the reasons for not performing one before study initiation.

4. Thank you for including your ethics statement: 

"TThe present study was conducted with the approval of the Research Ethics Committee of the organization to which the researchers belong (260024-1, 201700953).".   

For additional information about PLOS ONE ethical requirements for human subjects research, please refer to " ext-link-type="uri" xlink:type="simple">http://journals.plos.org/plosone/s/submission-guidelines#loc-human-subjects-research."

5. Thank you for stating in your Funding Statement:

'This work was supported in part by the grants-in-aid from the Japanese Ministry of Education, Culture, Sports, Science and Technology, Tokyo, Japan (Grants-in-aid for Nakayama, #23792624).

The funders had no role in study design, data collection and analysis, decision to publish, or preparation of the manuscript.'

a. Please provide an amended statement that declares *all* the funding or sources of support (whether external or internal to your organization) received during this study, as detailed online in our guide for authors at http://journals.plos.org/plosone/s/submit-now

Please also include the statement “There was no additional external funding received for this study.” in your updated Funding Statement.

6. Please include captions for your Supporting Information files at the end of your manuscript, and update any in-text citations to match accordingly. Please see our Supporting Information guidelines for more information: http://journals.plos.org/plosone/s/supporting-information

Reviewers' comments:

Reviewer's Responses to Questions

**Comments to the Author**

1. Is the manuscript technically sound, and do the data support the conclusions?

Reviewer #1: Yes

Reviewer #2: Yes

2. Has the statistical analysis been performed appropriately and rigorously? 

Reviewer #1: Yes

Reviewer #2: Yes

3. Have the authors made all data underlying the findings in their manuscript fully available?

Reviewer #1: Yes

Reviewer #2: Yes

4. Is the manuscript presented in an intelligible fashion and written in standard English?

Reviewer #1: Yes

Reviewer #2: Yes

5. Review Comments to the Author

Reviewer #1: Dear the authors of the manuscript entitled "Decreased Continuous Sitting Time Increases Heart Rate Variability in Patients with Cardiovascular Risk Factors"

Thank you very much for writing this manuscript which describes a valuable experiment discussing the positive effects of decreased sitting behavior on the outcomes of patients with cardiovascular risk factors

I have 2 points to mention here:

1. There was no mention on echocardiographic assessment for the patients included, which I think is relevant to the outcomes for these patients

2. The discussion part is deviated from the main theme of the study, please elaborate more about the results you obtained in the experiment ie(HF HRV and change in sitting behavior)

Otherwise I have no concerns about it

Thank you

Reviewer #2: This was an interesting study explaining the association between high-frequency heart rate variability and daytime sitting time among patients with cardiovascular risk factors. However, the authors should consider addressing the following:

1. The authors should provider how they arrived at the sample (53) of participants that participated in their study. Were the participants selected at random?

2. Under the Ethical consideration (page 7, lines 123-127) the authors need to specify the institutions that granted the ethical approval(s) for their study and the corresponding reference numbers. The authors are affiliated to five (5) institutions and this needs to be specified.

3. On page 8, line 151, the authors should replace the word “deceased” with “decreased”.

4. On page 13, the authors should consider providing a limitation of the methods used in their study. The authors also mentioned on lines 215 - 216 that “Patients who participated in this trial suffered from mild hypertension and/or stable angina pectoris.” The authors need to specify whether or not they conducted a trial and if so which type? In addition, was the trial registered? and was the protocol number provided anywhere in the methods?

6. PLOS authors have the option to publish the peer review history of their article (what does this mean?). If published, this will include your full peer review and any attached files.

Reviewer #1: **Yes: **salah eldien Altarabsheh

Reviewer #2: No

---

## [Author Response · Author response to Decision Letter 0]

16 May 2021

We wish to express our sincere appreciation to the reviewers for their insightful comments on our paper. The comments have helped us significantly in improving our paper.

We responded to each reviewer's questions and comments. We have uploaded the reviewer response's word file, so please check.

---

## [Decision Letter · Decision Letter 1]

4 Jun 2021

Decreased Continuous Sitting Time Increases Heart Rate Variability in Patients with Cardiovascular Risk Factors

PONE-D-21-08345R1

Dear Dr. Natsuki Nakayama,

We’re pleased to inform you that your manuscript has been judged scientifically suitable for publication and will be formally accepted for publication once it meets all outstanding technical requirements.

Kind regards,

Sharon Mary Brownie

Academic Editor

PLOS ONE

Reviewers' comments:

Reviewer's Responses to Questions

**Comments to the Author**

1. If the authors have adequately addressed your comments raised in a previous round of review and you feel that this manuscript is now acceptable for publication, you may indicate that here to bypass the “Comments to the Author” section, enter your conflict of interest statement in the “Confidential to Editor” section, and submit your "Accept" recommendation.

Reviewer #1: All comments have been addressed

Reviewer #2: All comments have been addressed

2. Is the manuscript technically sound, and do the data support the conclusions?

Reviewer #1: Yes

Reviewer #2: Yes

3. Has the statistical analysis been performed appropriately and rigorously? 

Reviewer #1: Yes

Reviewer #2: Yes

4. Have the authors made all data underlying the findings in their manuscript fully available?

Reviewer #1: Yes

Reviewer #2: Yes

5. Is the manuscript presented in an intelligible fashion and written in standard English?

Reviewer #1: Yes

Reviewer #2: Yes

6. Review Comments to the Author

Reviewer #1: Dear the authors

I want to thank you for revising the manuscript and taking in consideration the reviewers comments

I have no concerns

Reviewer #2: The authors have addressed my previous comments and the manuscript has been strengthened further.

However, as a discretionary comment, on page 14, lines 229-230, the sentence that reads "Despite these findings, the present study has some limitations. Patients who participated in this analysis had mild hypertension and/or stable angina pectoris." I suggest the word 'analysis' should be substituted with 'study'. This is because the patients participated in the study and not the analysis.

Otherwise, all my comments have been duly addressed.

7. PLOS authors have the option to publish the peer review history of their article (what does this mean?). If published, this will include your full peer review and any attached files.

Reviewer #1: **Yes: **Salah Eldien altarabsheh

Reviewer #2: No

---

## [Editor Report · Acceptance letter]

8 Jun 2021

PONE-D-21-08345R1 

Decreased continuous sitting time increases heart rate variability in patients with cardiovascular risk factors 

Dear Dr. Nakayama:

I'm pleased to inform you that your manuscript has been deemed suitable for publication in PLOS ONE. Congratulations! Your manuscript is now with our production department. 

Kind regards, 

on behalf of

Professor Sharon Mary Brownie 

Academic Editor

PLOS ONE